# Anti-Flavivirus Vaccines: Review of the Present Situation and Perspectives of Subunit Vaccines Produced in *Escherichia coli*

**DOI:** 10.3390/vaccines8030492

**Published:** 2020-08-31

**Authors:** Sergio C. Araujo, Lennon R. Pereira, Rubens P. S. Alves, Robert Andreata-Santos, Alex I. Kanno, Luis Carlos S. Ferreira, Viviane M. Gonçalves

**Affiliations:** 1Laboratory of Vaccine Development, Instituto Butantan, São Paulo–SP 05503-900, Brazil; sergio.araujo@butantan.gov.br (S.C.A.); alex.kanno@butantan.gov.br (A.I.K.); 2Laboratory of Vaccine Development, Institute of Biomedical Sciences, Universidade de São Paulo, São Paulo–SP 05508-000, Brazil; lennon_rp@usp.br (L.R.P.); rpsa7@usp.br (R.P.S.A.); randreata@usp.br (R.A.-S.)

**Keywords:** flavivirus, mosquito-borne diseases, subunit vaccines

## Abstract

This article aims to review the present status of anti-flavivirus subunit vaccines, both those at the experimental stage and those already available for clinical use. Aspects regarding development of vaccines to Yellow Fever virus, (YFV), Dengue virus (DENV), West Nile virus (WNV), Zika virus (ZIKV), and Japanese encephalitis virus (JEV) are highlighted, with particular emphasis on purified recombinant proteins generated in bacterial cells. Currently licensed anti-flavivirus vaccines are based on inactivated, attenuated, or virus-vector vaccines. However, technological advances in the generation of recombinant antigens with preserved structural and immunological determinants reveal new possibilities for the development of recombinant protein-based vaccine formulations for clinical testing. Furthermore, novel proposals for multi-epitope vaccines and the discovery of new adjuvants and delivery systems that enhance and/or modulate immune responses can pave the way for the development of successful subunit vaccines. Nonetheless, advances in this field require high investments that will probably not raise interest from private pharmaceutical companies and, therefore, will require support by international philanthropic organizations and governments of the countries more severely stricken by these viruses.

## 1. Introduction

Flaviviruses are enveloped and positive-sense single-stranded RNA viruses of the *Flavivirus* genus and Flaviviridae family. Most of them are transmitted to their hosts by hematophagous mosquitoes or ticks. However, alternative transmission routes in humans, such as sexual and transplacental transmission, have been described. The distinguishing characteristic of the *Flavivirus* genus is the type I cap (m^7^ GpppAmp) at the 5′-end of the genome, which is not seen in viruses of the other genera [1]. The yellow fever virus (YFV) is the prototype of the family, which encompasses other species of viruses that cause important human diseases, such as Dengue virus (DENV), West Nile virus (WNV), Zika virus (ZIKV), and Japanese encephalitis virus (JEV).

Diseases caused by flaviviruses have great health and socioeconomic burden to countries mainly located at tropical and subtropical regions. More than 3 billion people are at risk of JEV infection. DENV infects approximately 390 million people annually. WNV is the most geographically widespread flavivirus since it also affects countries in the Northern hemisphere. The last ZIKV outbreak alarmed the world due to the congenital Zika syndrome, which includes microcephaly. Despite the existence of a potent vaccine, YFV has re-emerged as a threat to public health in recent years. Moreover, factors such as climate changes, increased human migration, and the spreading of mosquito vectors have raised concerns over the introduction of these viruses into new environments. Therefore, the development of new vaccines and/or novel manufacturing techniques to rapidly produce large amounts of flavivirus antigens is urgent. This article is a review of (1) the proposed antigens for use in the development of subunit vaccines for DENV, ZIKV, WNV, YFV, and JEV; (2) the recombinant platforms that have been used to produce these vaccines; and (3) the potential advantages and limitations of producing these antigens in *Escherichia coli* systems.

## 2. Flavivirus Structure and Proteins

All flaviviruses have the following three features in common: (1) identical genome organization, (2) similar polyprotein processing, and (3) tridimensional structure (Figure 1). Each contains a single-stranded positive RNA, which codifies a single polyprotein. Successive cleavages of the single polyprotein, by both viral and cellular proteases, generate three structural proteins: capsid (C), pre-membrane (prM), and envelope (E) proteins. Seven nonstructural (NS) proteins are also generated—NS1, NS2A, NS2B, NS3, NS4A, NS4B, and NS5. The virion structures of flaviviruses were determined mostly by cryo-electron microscopy [2,3,4,5,6].

The virion envelope shows an icosahedral symmetry in which envelope (E) protein dimers are arranged in a herringbone manner [1]. Immature virions become mature when prM proteins are processed in vivo, causing a conformational change on the virus surface, from a bumpy, looser surface to a smooth, compact surface, in which E proteins tightly interact with one another to form three sets of dimers lying parallel to each other and forming a raft [2,3]. Recent work on ZIKV and DENV shows the induction of non-spherical, club-shaped, or caterpillar-shaped morphologies at temperatures ≥37 °C. These morphologies were associated with the scape of the virus from the immune system [7].

The E protein of flaviviruses is responsible for the interaction with the host cell receptor, which triggers virus internalization. When the virus enters the endosome, the low pH induces conformational changes. These changes expose the fusion loop of E protein and lead to the fusion of the viral protein with endosomal membranes. Next, the RNA of the virus is released and the translation of the viral polyprotein starts. The polyprotein is cleaved to generate structural and NS proteins. The NS proteins act in genome replication. Newly synthesized RNA and C protein are packaged by prM and E to assemble immature virus particles. These particles bud into the endoplasmic reticulum and are glycosylated. Finally, the immature virion is transported through the trans-Golgi network, where prM is cleaved by the protease furin into pr peptide and M protein, and the trimeric prM-E are rearranged to dimeric M-E heterodimers, thus forming the smooth mature virion particles, which are excreted to infect other cells [1,8].

Each nonstructural flavivirus protein has a function in virus replication. The structure similarity shared by these proteins from different flaviviruses was recently analyzed [8]. NS1 is a multifunctional protein that has two forms: (1) a cell-associated form that acts in viral RNA replication and as cofactor for virus infection and (2) a secreted form that regulates the innate immune response [9]. In some flaviviruses, a-1 ribosomal frameshift event produces also a NS1′ protein, and mutations that abolished NS1′ production in JEV led to reduced viral neuroinvasiveness [10]. NS2A is small protein reported to be involved in viral RNA replication [11,12], modulation of the host-antiviral interferon response [13,14,15,16] and virus particle assembly/secretion [17,18,19]. NS2B acts as a cofactor to NS3 protease domain, assisting its folding and catalytic activity [8,20]. NS3 has two domains: N-terminal protease and C-terminal helicase. The NS3 protease domain is a chymotrypsin-type serine protease [21]. The NS3 helicase domain presents helicase and nucleoside 5′-triphosphatase activities [22]. NS4A and NS4B have multiple functions involving viral replication and virus–host interactions. The reported functions of NS4A involve endoplasmic reticulum membrane rearrangement [23], participation in virus replication complexes formations [24], autophagy induction to prevent cell death and help viral replication [25] and regulation of NS3 helicase ATPase activity [26]. NS4B is reported to interact with the NS3 helicase domain and dissociate it from single-strand RNA [27]. NS4B can induce the unfolded protein response in the host cells and inhibit interferon (IFN) signaling [16,28,29]. NS5 is the largest NS protein directly involved in capping and RNA replication. It is also involved in interferon suppression and has two domains: an N-terminal S-adenosylmethionine-dependent methyltransferase domain, which adds the 5′-RNA cap to assist polyprotein translation and diminish activation of host innate-immune responses, and a C-terminal RdRp region, which is involved in the RNA-replication process [30,31,32].

The E protein is a natural candidate for subunit vaccines, since it is on the virus surface and plays a direct role on host cell receptor binding and cell fusion. The ectodomain (soluble N-terminal region) of E monomer has three domains: a beta-barrel domain I (EDI), a finger-like dimerization domain II (EDII) that contains a fusion loop, and an immunoglobulin-like domain III (EDIII), which contains the receptor-binding site and the major type-specific neutralization epitopes; consequently, the majority of subunit vaccine candidates uses E protein or EDIII as antigen [33,34]. Nonetheless, cellular immune response can also be protective for flavivirus and in some cases is required in order to generate robust protection. For these reasons, there are some proposals for subunit vaccines that employ NS1, NS3, and NS5 as vaccine antigens [35,36,37,38,39,40,41]. Since subunit vaccines are safer than virus attenuated vaccines (but are less immunogenic), they could be the preferred antigen candidates for specific risk groups such as young children, the elderly, and immunocompromised persons. Subunit vaccines could also be employed as a safer strategy for prime-boost immunization regimens that combine live-attenuated vaccines and subunit vaccines.

## 3. Yellow Fever Virus

Yellow fever (YF) is an acute disease that affects humans and non-human primates (NHPs) and is caused by the Yellow Fever virus (YFV). Clinical manifestations of YF vary from asymptomatic individuals to a systemic viral sepsis with viremia; fever; prostration; liver, kidney and heart injury, hemorrhage; and shock, which can result in death [42]. YFV remains endemic or enzootic in many South American and African countries, recurrently causing outbreaks and epidemics [43,44]. In Brazil, from 1967 to 1999, sylvatic human YF outbreaks were usually reported mostly in the Amazon Basin in the north and mid-west regions, and fewer cases were reported in the Southeast region [45]. After 1999, however, most reported cases occurred outside of the Amazon Basin, mainly in southeast, mid-west, and south regions of Brazil [45]. From 2016–2019, massive sylvatic YF outbreaks occurred in states of Minas Gerais, São Paulo, Espirito Santo, Rio de Janeiro, and Bahia, affecting both humans and NHPs [46,47,48]. During this period, the number of confirmed cases and deaths due YF was respectively 2.82 and 1.57 times higher than the sum of all cases and deaths reported in the prior 36 years (1980–2015) [44]. This evidence highlights that even with licensed and efficient vaccine available, government commitment in disease surveillance is essential to avoid outbreaks.

The YF live-attenuated vaccine (YFV-17D) was developed in 1936 using the 17D strain. All YF vaccines produced today are derived from this strain. This vaccine is produced in embryonated chicken eggs and the techniques applied to vaccine manufacture have changed little since its development in 1940s [49]. Even though the protection mechanisms are not yet fully elucidated and the manufacture process is presumed to be outdated, the YFV-17D is an efficient and quite safe vaccine [50]. YFV-17D had been considered the safest licensed vaccine, but this changed when some severe adverse events related to vaccination were discovered in 1996. A few cases of YF-vaccine-associated viscerotropic disease (YEL-AVD) and YF-vaccine-associated neurotropic disease (YEL-AND) have been reported since then. These events have greater incidence in elderly people (over 60 years old). Fortunately, these events are rare. For example, in the United States, these events were estimated at 0.4 cases per 100,000 vaccinated subjects [45,51]. The adaptive immune responses to YFV-17D are fast, robust, and durable (usually life-long). Studies show that neutralizing antibodies induced after vaccination with YF-17D target a low number of conserved epitopes in the E protein, and antibody titers reach as high as 30 times the needed value for protection after vaccination [52,53]. Despite antibodies being considered the foremost mediator for YF protection, cellular immune response also plays an important hole. After immunization, CD4^+^ and CD8^+^ T cells appear in the bloodstream, differentiate, and remain there for the long term as memory T cells [45]. Another central feature of the YF-17D vaccine is its capacity to induce robust innate immune responses including production of interferons, activation of inflammasome, and activation of complement elements. These strong, fast, and integrated innate response pathways are the reason for the robust and durable induced cellular and humoral immune responses [54,55]. Because of these features, the YF-17D vaccine is recognized as one of the most effective vaccine ever created [56].

Due to excellent immune properties, the YFV-17D vaccine has been used as a vector for expressing epitopes of other flaviviruses, such as JEV [57], DENV [58], WNV [59], and ZIKV [60], and of antigens from other pathogens, such as malaria [61] and HIV [62]. Despite their demonstrated efficacy, YFV-17D-based vaccines are not recommended for elderly over 60 years old, infants younger than six months, pregnant and breastfeeding women, people who are immunocompromised, and individuals with egg-associated hypersensitivity [63,64,65,66,67,68]. Therefore, the development an alternative vaccine, one that is safe for all people regardless of age and health conditions and capable of generating durable and robust protection, continues to be a challenge.

Recombinant YFV proteins were mainly generated with sole purpose of structural characterization (Table 1). Recombinant EDIII with an N-terminal His-tag was obtained as inclusion bodies in *E. coli* using the pET-15b [34] or pET-20b vectors under control of the *pel*B signal sequence to be exported to the periplasm [69]. Recombinant capsid protein lacking the C-terminal hydrophobic sequence was produced in a soluble form by using *E. coli* strain BL21(DE3) RIL [70]. NS2A, NS2B, and NS4B were obtained in recombinant *E. coli* strains in order to identify the cleavage sites [11], while recombinant NS3 was produced to characterize the enzymatic activities of the protein [71,72]. Recombinant NS5 was produced in HEK293T cells for phosphorylation characterization [73,74]. Recombinant YFV NS1 was obtained in *E. coli* strain Lemo21 (DE3) using vector pBT7-N-His [75]. NS1 and E proteins were also produced in insect and mammalian cells [76]. Except for two previous studies, none of these recombinant proteins were tested as potential vaccine antigens under experimental conditions. Immunizations performed with chimeric YFV NS1-β-galactosidase, produced in *E. coli*, induced NS1-specific antibodies and conferred protection to mice [35]. In another study, a recombinant YFV E protein produced in transgenic plants was used to immunize mice and monkeys. The immunization elicited neutralizing antibodies, and most of the vaccinated mice were protected against lethal challenge with the virus [77]. With the purpose of developing a safer and alternative YF vaccine, Tosta et al. applied bioinformatics tools to design a multi-epitope YF vaccine based on the genome sequence of 137 YFV strains [78]. They proposed to reduce production costs by producing the recombinant protein in *E. coli*. However, the efficacy of such a vaccine still awaits experimental validation.

## 4. Japanese Encephalitis Virus

Japanese encephalitis (JE) is the most prevalent viral encephalitis in more than 20 countries in South Asia and in the Western Pacific Region and is known as “Orient’s plague” [79]. In fact, over 3 billion people (~43% of the human population) are at risk of infection. Annually, there are approximately 68,000 cases each year and 20,000 deaths from JE. Most JE virus (JEV) infections are asymptomatic or mild, with symptoms such as moderate headache and fever. However, 0.4% of the cases evolve into a severe clinical form of JE characterized by high fever, headache, disorientation, seizures, and coma. About 30% of severe infections are fatal, and up to 30% of those who survive a severe infection will suffer permanent neurological sequelae [80].

The JE virus was isolated in 1924. It was described as a small icosahedral-enveloped arbovirus transmitted by *Culex* mosquitoes [81]. Based on the nucleotide homology of E gene, JEV was categorized into five JEV genotypes: GI, GII, GIII, GIV and GV, though all of these genotypes belong to the same serotype. This is evidenced by the absence of secondary infections, in vitro seroneutralization tests of heterologous genotypes, and the fact that vaccination has significantly decreased the burden of the disease irrespective of the genotype [82,83,84].

The protection after JEV vaccination correlates with the level of neutralizing antibodies produced. Several reports demonstrated that a protective status is reached when serum from vaccinated individuals presents neutralization titers in plaque reduction neutralization test (PRNT50) of at least 10 [82,85,86], a rather low PRNT50 titer value if compared with other viral diseases [87,88,89]. JEV vaccine-induced T cell responses are observed in live attenuated vaccines only; and they preferentially target the NS3 protein. In addition, the generation of CD8^+^ T cell responses is better indicator of protective immunity than that of CD4^+^ T cell responses.

Despite the efficacy of vaccination, the use of several JEV vaccine formulations approved for humans has been discouraged or discontinued for a variety of reasons. Mouse brain-derived inactivated (MBDI) vaccines were the first to be approved and implemented for large-scale human immunization [90]. The Nakayama JEV strain was the first to be approved [91] and was later replaced by the Beijing-1 JEV strain [92]. However, it was discontinued in 2005 due to its requirement of multiple doses, limited immunogenicity, and adverse symptoms, especially in Caucasians. Cell-culture-derived inactivated (CCDI) vaccines replaced MBDI vaccines because their production process is more easily controlled. The development of several CCDI vaccines (especially using VERO cells) has been reported [93,94,95]. Virus attenuation attempts led to the development of live-attenuated vaccines (LAVs), with the subsequent approval of the SA-14-14-2 LAV formulation by several national governments [96]. The SA-14-14-2 LAV vaccine impaired vector transmission and showed greater immunogenicity than inactivated vaccines. However, it fell into disuse due to concerns about its virulence rebound and the threat that it posed to swine, which are frequently raised in JEV-endemic areas. Vector-based vaccines have also been tested against JEV infections, but only the JE-CV construction, which is based on the YFV-17D backbone, is licensed for human use. The NYVAC-JEV (attenuated vaccinia virus backbone) and the ALVAC-JEV (canarypox virus backbone) vaccines showed lower immunogenicity or raised safety concerns during clinical trials [97]. Moreover, in a comparison with the SA-14-14-2 LAV vaccine, the JE-CV vaccine candidate showed equivalency when tested in a child cohort [98]. Subunit vaccines were also evaluated in preclinical conditions, though they did not advance to clinical trials. 

JEV subunit vaccines were produced for preclinical evaluations using the *E. coli* platform (Table 2). Initial studies demonstrated that a minimum structural 95-mer antigenic domain within the E glycoprotein was capable of binding neutralizing monoclonal antibodies (mAb). However, immunization with this E fragment did not induce neutralizing immune responses in mice. Its inability to induce neutralizing immune responses in mice was probably due to the loss of antigenic epitopes after refolding of the protein from the insoluble fraction [99]. Two fragments of the JEV E glycoprotein, Ea (N-terminal) and Eb (C-terminal), were produced in *E. coli.* Whereas the Eb fragment was obtained in a soluble form, the Ea fragment was recovered from the insoluble fraction and refolded. Both recombinant fragments were recognized by JEV neutralizing mAb and hyperimmune mouse serum, but only mice immunized with two doses of Eb fragment administered with the Freund’s adjuvant induced high neutralizing antibody titers and achieved partial protection. This evidence highlights the importance of proper protein folding and generation of efficient neutralizing antibodies [100]. Later, the in vivo expression of Ea in mouse immunized with a DNA vaccine was also found to be non-protective, unless it was preceded by the signal peptide of the gene of M protein, revealing the importance of this peptide to Ea proper folding, immune activity, and induction of protection [101].

The receptor-binding domain III of E protein (EDIII) from attenuated JEV CH2195LA isolate was fused with thioredoxin to produce the protein in a soluble form [102]. The immunization of mice with this protein, which was associated with different adjuvants, induced neutralizing antibodies and protective immunity [102]. In another study, the antigenicity, immunogenicity, and protective immunity induced by JEV vaccines containing EDIII and NS1 were evaluated [36]. Mice immunized with these recombinant proteins elicited antigen-specific antibody responses, but virus-neutralizing antibodies were induced only in mice immunized with EDIII. Surprisingly, the NS1-containing vaccine formulation induced high serum anti-NS1 IgG1 titers, which correlated with greater survival rates (87.5% for NS1; 62.5% for EDIII) [100]. Although the generation of neutralizing antibodies to structural virus proteins is considered the main protection correlate of JEV vaccines, these results demonstrated that such a correlation is not complete. Taken together, these results emphasize the perspectives of alternative JEV vaccines based on recombinant proteins produced in *E. coli.*

## 5. West Nile Virus

West Nile virus (WNV) is a neuroinvasive and neurotropic virus. While most of the infected people are asymptomatic, around 20% will develop West Nile fever, a condition characterized by a sudden onset of headache, fever, myalgia, fatigue, and vomiting. Patients can recover within a week or they can remain in a debilitating state for months. Around one out of 150 patients infected with WNV will develop neurological symptoms such as meningitis, encephalitis, acute flaccid paralysis, and various ocular manifestations. West Nile fever is particularly serious in older adults, who usually show more severe clinical manifestations [103,104,105].

WNV was first isolated in 1937 in the West Nile district in Uganda [106], but today, it can be found in other parts of Africa, the Americas, Europe, and Oceania. It is the most widespread arbovirus in terms of geographic distribution. While case reports are sparse, outbreaks are recurrent. One of the latest outbreaks occurred in 2018 and involved over 2000 human cases across 15 countries in Europe [107]. Since its introduction into the US in 1999, almost 25,000 cases of neurological disease and 2300 deaths were reported, making WNV the leading cause of an epidemic of mosquito-borne encephalitis [108]. WNV represents such a serious problem that, along with ZIKV, is the only mosquito-borne pathogen that is tested in blood transfusions in the US [109].

Though there are veterinary vaccines licensed for use in susceptible animals, mainly horses, no licensed human vaccine for WNV exists. The veterinary vaccines include inactivated whole-virus vaccines (WEST-NILE INNOVATOR by Zoetis (Parsippany, NJ, USA) and Vetera WNV by Boehringer Ingelheim Vetmedica (Duluth, GA, USA), a non-replicating vector based on live canarypox virus (Recombitek Equine WNV by Boehringer Ingelheim Vetmedica (Duluth, GA, USA), and an inactivated chimera flavivirus vaccine (Equi-Nile by Merck Animal Health (Madison, NJ, USA). These vaccines confer immunity for one year only, so revaccination is recommended. Possibly, due to less regulatory stringency, veterinary vaccines can move more quickly though clinical testing than those for human use. Other WNV veterinary vaccines were licensed but are no longer available. These include the DNA vaccine WEST NILE-INNOVATOR DNA and the live attenuated chimeric vaccine PreveNile [110].

Despite the potential threat of epidemic outbreaks, bringing a human WNV vaccine to phase III clinical trials involves high costs and a rather reduced market value. A study published in 2006 indicated that “universal vaccination against WNV disease would be unlikely to result in societal monetary savings” [111]. Even 20 years after the introduction of WNV into the US, a recent study concluded that it would be more cost-effective to invest in an age-based vaccination program as opposed to a universal one [112]. Therefore, such vaccination programs do not interest large pharmaceutical companies. The US National Institute of Allergy and Infectious Diseases (NIAID) supports the research and testing of a variety of WNV vaccine candidates including inactivated virus, peptides, attenuated, and chimeric virus vaccines. This research is mainly conducted by universities and biotech companies [113], which usually face budgetary, academic bureaucracy, and marketing-related challenges.

The epidemiological profile of WNV presents another hurdle to the advancement of efficient WNV vaccines to the clinical trial. The uncertainty of where and when WNV will appear and the fact that it causes outbreaks at different scales makes it is hard to get approval for and set up phase III clinical trials. The concomitant occurrence of other flaviviruses may have implications for the approval and set-up of phase III clinical trials. Cross-reactivity of antibodies and the possibility of the antibody-dependent enhancement (ADE) emergence are additional factors to be considered when deciding to launch such trials. Additionally, the differences between the epitope-specific profile of humans and mice may make the translation of mice results to clinical conditions difficult [114].

Human clinical trials of WNV vaccines have been done using DNA-based and vector-based vaccines, inactivated virus vaccines, and subunit vaccines. Initial phase I/II trials of formaldehyde-inactivated WNV, live attenuated YFV or DENV vectors expressing WNV prM/E proteins, DNA plasmid carrying prM/E proteins, and recombinant E protein vaccine candidates demonstrated promising results for the induction of neutralizing antibodies in most of the enrolled participants [115,116]. In addition, no adverse effects were reported. Despite these promising results, none of them progressed to phase III. The recombinant E protein fragment WN-80E is the only subunit vaccine that entered clinical trials and was produced in *Drosophila* cells [117]. This vaccine induced the generation of neutralizing antibodies after three doses in a dose-dependent manner [118]. A key aspect of this vaccine candidate is the exclusion of the transmembrane region of the antigen (E protein). This deletion resulted in the expression of soluble protein secreted to the extracellular medium. Moreover, unpublished data claim that the antigen is properly glycosylated and maintains the native conformation of the viral protein [118].

Subunit vaccines based on recombinant proteins expressed in *E. coli* were developed and tested at a preclinical level (Table 3), but they did not advance to clinical trials. Viral proteins are commonly produced as inclusion bodies in *E. coli*. The recombinant ectodomain of WNV E protein was formulated with the particulate saponin-based adjuvant Matrix-MTM. It induced neutralizing antibodies and 100% protection in mice [119]. Likewise, the E protein domain III (EDIII), alone [120] or fused/conjugated to other proteins [121,122] and formulated with adjuvants, induced protective immunity against lethal challenge in mice. All these data demonstrate the potential for the development of WNV vaccines by using formulations of recombinant proteins produced in *E. coli*.

Although it is difficult to establish a single explanation for their absence in clinical trials, certain issues may have delayed the evolution of preclinical studies of WNV vaccine candidates produced in *E. coli.* These include difficulties with achieving the correct folding of proteins from inclusion bodies (IBs) and determining an appropriate adjuvant. Early studies showed that the immunogenicity of WNV antigens is greatly affected by conformation. Also, disulfide bond patterns differ when proteins are produced in insect cells or *E. coli* [132]. Several WNV vaccine candidates employed proteins purified from IBs, and they were evaluated with different adjuvants to obtain the best immune response (Table 3). Additionally, post-translational protein modifications, such as glycosylation, may also affect the immunogenicity of WNV proteins. For example, only the glycosylated WNV E protein can optimally bind to C-type lectin DC-SIGN(R), a receptor that can either enhance the infection [133] or promote immune cell activation and antigen uptake [134]. Although bacterial cells do not usually promote such post-translational modifications, novel genetically modified *E. coli* strains that perform the glycosylation of recombinant proteins are good alternatives to deal with this problem [135]. Culturing conditions, solubility tags, and different expression vectors and strains can easily be evaluated in order to obtain a soluble antigen [136].

With the increasing number of options available, *E. coli* can become a suitable host for the production of protein-based WNV vaccines for preclinical evaluations and, later, progress to clinical trials. The incorporation of adjuvants and/or combinations with alternative immunization strategies, such as prime-boost regimens may contribute to overcome the low immunogenicity of recombinant proteins and improve immune responses.

## 6. Dengue Virus

Annually, dengue virus (DENV) infects approximately 390 million people worldwide. This results in about 500,000 hospitalizations and a death toll of approximately 20,000 lives each year [137]. There are four serotypes of DENV capable of infecting humans. They are distributed among more than 100 countries in tropical and subtropical regions around the world [138]. Most part of DENV infections (about 75%) are asymptomatic and self-limited. However, symptomatic infections can show different degrees of illness. The World Health Organization (WHO) classifies DENV infections as DF for asymptomatic and/or dengue fever, DWS for a more severe dengue with warning signs, and SD for severe dengue with clinical complications [139,140]. Patients with DWS can present fever (37.5–38 °C), headache, retro-orbital and abdominal pains, myalgia, arthralgia, a rash, flushing of the face, anorexia, and nausea. In addition to these symptoms, SD patients can also have plasma leakage, fluid accumulation, severe bleeding, and organ impairment that can be fatal [139]. Although the determinants of the complications related to SD are not well understood, variables such as secondary infections with different serotypes, immunity, age, and individual genetics can affect the risk of developing the more severe forms of the disease [141,142,143]. 

There is not an effective therapy available for dengue. Current treatments address the symptoms of the disease. This includes fluid replacement and blood transfusion, which require hospitalization [144]. There is, however, a prophylactic tetravalent live-attenuated virus vaccine, Dengvaxia^®^, also known as CYD-TDV, that was developed by Sanofi Pasteur and is licensed in approximately 20 countries. CYD-TDV consists of four chimeric viruses based on the 17D strain of YFV. The sequences encoding the pre-membrane (prM) protein and the envelope glycoprotein (E) were replaced by those of each of the four DENV serotypes. Despite being licensed, however, recent studies have shown that this vaccine may present risks to individuals without prior immunity to DENV. This can result in an increase in the number of severe cases after primary DENV infection and, consequently, in the number of hospitalizations [140,145,146]. Additionally, according to WHO guidelines, CYD-TDV should not be applied to the general population. Rather, it should only be used in those countries with demonstrated seroprevalence (≥80%) and in individuals between nine and 45 years of age [143,147]. Thus, the search for effective and safe DENV vaccines is still necessary.

During the past few years, another six DENV vaccine candidates advanced to clinical studies (reviewed by Pinheiro-Michelsen, et al. [148]). Among them, two are in phase III trials. These are LATV by NIAID (Bethesda, MD, USA) and the Butantan Institute (São Paulo, SP, Brazil) and TAK-003 by Takeda Pharmaceutical Company Limited (Tokyo, Japan). Both are based on the live-attenuated DENV and are capable to induce neutralizing antibodies and cellular immune response to all DENV serotypes [149,150,151,152,153]. Recently, TAK-003 demonstrated a promising efficacy of 80.2% in a phase III study [154]. Although data related to the performance of LATV vaccine in phase III trial is not yet available, this vaccine exhibited protective results in a human challenge model with the DENV 2 strain [5]. 

The main challenge to the development of an effective DENV vaccine is the need for a vaccine that induces a safe, long-lasting, and balanced immune responses for all four DENV serotypes. Such concerns are particularly relevant for formulations containing structural proteins, due to the risk of they present of increasing of the severity of the disease by antibody-dependent enhancement (ADE) [155]. This phenomenon occurs when the infectivity of the virus is increased in the cells that express Fc receptors when the antibodies binding to DENV but do not have neutralizing capacity or are present in sub-neutralization amounts [156,157]. Indeed, during CYD-TDV vaccine phase II and III trials, variable serotype-specific protective responses were reported, with the lowest response reported for DENV 2 (Serotype 1, 50–50.3%; Serotype 2, 9.2–35%; Serotype 3, 9.2–35%; Serotype 3, 74.0–78.4%; Serotype 4, 75.3–77.7%). This is indicative of an unbalanced immune response [145,158], and may represent a limitation of the CYD-TDV vaccine, especially among DENV-naive individuals [140].

Evidences indicate that induction of high titers of neutralizing antibodies does not signify a complete protection correlation. In contrast, certain aspects of T cell–mediated immunity, CD8^+^ T cells in particular, may be related to protection against DENV infection and disease [159,160,161]. Since the non-structural (NS) proteins are the major targets of the anti-DENV CD8^+^ T cell responses [162], vaccines that lack responses targeting NS proteins, such as the CYD-TDV vaccine, fall behind vaccines that induce responses to all DENV proteins, such as the LATV vaccine. Despite its capacity to induce cellular responses against the NS proteins, when administered in a tetravalent formulation, the NS responses shift from a heterogeneous response between NS3 and NS5 proteins to a response mainly targeting the NS5 protein. This is likely due to antigen accumulation, since the NS5 proteins are the most conserved NS protein among DENV serotypes [162,163,164]. Similar difficulties also occur during the development of humoral responses induced by attenuated vaccines. These difficulties may be avoided by the use of recombinant antigens. 

Over 30 DENV vaccine candidates based on recombinant antigens have been developed to date. However, the majority of these vaccine candidates have not advanced to clinical validation despite their promising preclinical results, as shown in Table 4. The tetravalent vaccine V180, developed by Merck & Co. (Kenilworth, NJ, USA), is the only one that is presently under clinical testing. The vaccine is based on truncated versions (DEN-80E) of the E protein from DENV1-4, produced in Drosophila Schneider-2 (S2) cells, and uses the adjuvant ISCOMATRIX^TM^ (CSL Behring, King of Prussia, PA, USA) [165]. Preclinical tests using mice and NHPs showed the induction of Th1 biased cellular immune responses, the presence of neutralizing antibodies and protective immunity against lethal challenge. In humans, reports indicate that V180 induce neutralizing antibodies, with 85.7% seroconversion [165,166,167]. On other hand, among the different DENV subunit vaccines tested until now, those based on recombinant proteins produced in *E. coli* strains have been intensively evaluated (Table 4). This begs the question, why haven’t DENV formulations based on this platform not advanced to clinical trials?

The recombinant DENV antigens produced in *E. coli* cells include full length or fragments of different proteins, fused or not with other proteins (chimeras), which were associated with different adjuvants. Most part of the studies are based on the structural proteins (E and C) but non-structural proteins, such as NS1, NS3, and NS5, were also investigated (Table 4). The most advanced strategies use chimeric proteins to improve the immunogenicity of the DENV proteins. Tetra DIIIC vaccine, which is composed by chimeric forms of the E protein domain III (EDIII) fused to the capsid (C) protein from all DENV serotypes, has been tested with different adjuvants and form protein aggregates when associated with oligodeoxynucleotides. These formulations induced neutralizing antibodies, activation of B and T cells, and protective immunity in mice and monkeys [177,179,180,181]. Additionally, the chimeric protein also boosted the natural immune responses generated after DENV infection in monkeys [178] or in a heterologous prime-boost regimen with the live-attenuated LATV vaccine [182]. Along these lines, chimeric proteins using the E protein domain b (aa 298–400) from DENV 1–4 and fused with maltose binding protein (MBP) induced neutralizing antibodies in mice and NHP. However, protective immunity was registered only in mice [173,174,175,176]. Others monovalent or tetravalent formulations based on full E protein or EDIII were also evaluated in mice, and some of them induced neutralizing antibodies (Table 4). 

Recently, the role played by recombinant DENV NS protein produced in *E. coli* has been evaluated. Formulations using the NS1 protein associated with a derivative of the heat labile toxin (LTG33D) or Freund adjuvants were capable to stimulate humoral and cellular immune responses in mice, inducing partial protection to DENV2 [37,203]. NS5 protein was also protective to DENV2 strains, which was accompanied of enhanced antigen-specific antibodies and expansion of IFNy/TNF-α producing T cells, even in the absence of adjuvant [38]. Due to the presence of cytotoxic T cell epitopes in NS3, this antigen was recently tested in combination with a chimeric DENV/JEV NS1. Mice immunized with this vaccine showed activation of antigen-specific CD4^+^ and CD8^+^ T cells, elicited antibody responses, NS3-specific cytotoxic T lymphocyte (CTL) activities, and protective effects on a DENV2 infection model [40]. Despite the fact that the protective role of NS3 has been demonstrated only in vaccination platforms, such as DNA vaccines [41], the recombinant protein produced in *E. coli* proved to be immunogenic in mice, leading to activation of antigen-specific antibody and cellular immune responses [39].

Taken together, these studies demonstrate that recombinant DENV proteins produced in *E. coli* are capable to induce protective immunity at experimental conditions and, thus, represent promising candidates for testing in clinical conditions. Nonetheless, most of the DENV recombinant proteins produced in *E. coli* showed reduced solubility and required the use of refolding techniques. This is specifically critical for structural proteins, such as the E protein, for which conformational epitopes are the main targets to neutralizing antibodies [204,205]. In addition, as usually observed with subunit vaccines, DENV vaccines based on recombinant proteins require the use of adjuvants to improve induction of antigen-specific immune responses. Such limitations may influence the development pipeline and translation to clinical trials.

## 7. Zika Virus

The 2015–2016 Zika outbreak in the Americas infected up to 73% of the population in some cities [206]. Before this, ZIKV infection was considered to be benign with trivial health consequences. However, during the last epidemics, ZIKV infection was related to nervous system diseases, including Guillain–Barré and congenital Zika syndromes, that, among other complications, resulted in thousands of cases of microcephaly. The clinical manifestations of congenital Zika syndrome are not yet fully elucidated. Many children born from infected mothers show no signs at birth, but they developed abnormal neurodevelopmental outcomes later in life [207,208]. Although the number of cases has decreased considerably since 2016, ZIKV is now endemic in tropical regions around the world. Five years since the last epidemics, there are still no licensed vaccines available [209].

Antibody responses are considered the most important indicator for an effective flavivirus vaccine. Indeed, monoclonal antibodies are capable of protecting mice from ZIKV infection [210], and T cell depletion after immunization does not interfere in this protection [211]. Unlikely DENV infections, in which CD4^+^ T cells target structural proteins and CD8^+^ T cells target mainly nonstructural proteins, in ZIKV infection, both CD4^+^ and CD8^+^ T cells target structural proteins [212,213]. The depletion of CD8^+^ T cells increased mortality in mice [214] and indicate that T cell responses also play an important role in protection. Even though neutralizing antibodies are considered the main contributor to protective immunity, the aforementioned evidences indicate that an optimal ZIKV vaccine should induce both cellular and humoral immune responses [181].

Several approaches have been taken in the pursuit of a ZIKV vaccine. Such approaches include RNA-based, DNA-based, and viral-vector-based vaccines, as well as attenuated viruses, inactivated viruses, and recombinant protein subunits. The formulations that have advanced to clinical trials use attenuated viruses, inactivated viruses, RNA-based, DNA-based, or viral vector vaccines (Table 5). To date, the only ZIKV vaccine that advanced to a phase II clinical trial is a DNA-based vaccine. It is called RC-ZKADNA090-00-VP (ClinicalTrials.gov Identifier: NCT03110770), and it encodes the prM and E proteins. During phase I, this vaccine did not show severe systemic reactions, but presented a few mild to moderate adverse effects. Moreover, all the participants (14/14) developed neutralizing antibodies. However efficacy, durability and protective effect in humans are not yet evaluated [215]. Results from a phase II trial are not yet available.

None of the ZIKV subunit vaccine advanced to clinical trials despite promising results in pre-clinical conditions (Table 6). Ham and collaborators have demonstrated that a recombinant subunit vaccine containing the N-terminus of the ZIKV envelope protein (E90) produced in *E. coli* induced robust and specific humoral responses in mice. The passive transfer of sera collected from immunized mice fully protected neonatal mice from a lethal virus challenge [216]. The same ZIKV vaccine was tested in mouse models for in utero and neonatal infection. Considerable reduction of brain cells infection was observed in fetus and suckling mice. Furthermore, this vaccine prevented the occurrence of microcephaly compared to unvaccinated controls mice. In the same study, the authors demonstrated that this vaccine protected mice even 140 days after immunization [217]. Despite its excellent results, the E90 subunit vaccine did not advance to non-human primate testing nor to clinical trials.

It is normally accepted that eukaryotic expression systems are the best at expressing viral antigens, particularly to induce neutralizing antibodies. However, only a few studies have actually compared the immunological performances of the eukaryotic and prokaryotic platforms producing the same protein. Liang et al. compared the immunogenicity and protective efficacy of ZIKV E protein (E80) produced in eukaryotic (*Drosophila* S2) and prokaryotic (*E. coli* BL21) cells [218]. The authors clearly demonstrated that there are not any significant differences in test results with regards to the immunogenicity and induction of protective immunity among mice immunized with antigens from both expression systems [218].

Along the same line, Amaral et al. compared a subunit vaccine based on the ZIKV E protein produced in *E. coli* (adjuvanted with poly (I:C) or CpG ODN) to a DNA vaccine (pVAX-E_ZIKV_) encoding the same antigen in homologous and heterologous prime-boost immunization regimens [219]. Their findings showed that mice immunized with one dose of E-poly (I:C) promoted a robust humoral response with a slightly increase after a booster dose and that mice immunized with E-CpG ODN required three doses to reach a similar antibody level. These results reinforce the importance of the role of adjuvant for the induction of appropriate immune responses. The homologous prime-boost immunization with E-poly (I:C) induced a stronger humoral response when compared to mice immunized with homologous pVAX-E_ZIKV_ or heterologous pVAX-E_ZIKV_ + E-poly (I:C) prime-boost regimens. The highest cellular response was detected in mice submitted to homologous E-poly (I:C) and heterologous pVAX-E_ZIKV_ + E-poly (I:C) prime-boost immunization regimens [219].

The lack of licensed, safe, and efficient ZIKV vaccine continues to be a public health concern. Even though a few vaccine candidates have advanced to clinical trials, they have not yet shown robust immune responses in humans. The *E. coli* expression platform is a simple and efficient approach to generate antigens, with particularly good results in pre-clinical conditions. This platform becomes even more attractive when one considers that ZIKV is now endemic and widespread across the globe. Yet, it has not advanced to NHPs or to clinical trials.

## 8. Discussion

Vaccine-induced immunity to flaviviruses is a reality, as demonstrated by the successful examples of the licensed YFV and JEV vaccines. These vaccines are based on live attenuated or inactivated virus, and their use in vaccination programs has generated significant public health benefits and reduced the incidence of the diseases [221,222]. Nonetheless, these vaccines show limitations for widespread use. The YFV-17D vaccine, for example, is contraindicated for use among young children (<6 months of age), pregnant women, immunocompromised individuals [223,224], or those over 60 years of age [63]. More recently, the licensed DENV vaccine Dengvaxia demonstrated unbalanced immune responses to the four DENV serotypes and an enhanced risk of severe disease cases, particularly among naive individuals. This has limited the recommended age range for the vaccine and restricted its use to countries with high DENV seroprevalence [140,143,145,147]. These facts demonstrate the need to improve the safety and efficacy of currently available flavivirus vaccines and the continuous demand for new vaccines for those diseases for which vaccines are not yet available. As presented in this review, subunit vaccines based on recombinant proteins produced in different platforms could be economically viable, safe, and effective alternatives for such unmatched technological challenges.

Recombinant subunit vaccines are known to be safe and easily customizable. Non-natural molecules composed of multi-epitopes identified by bioinformatics have been proposed as potential vaccine candidates, a strategy that still needs experimental validation [78,225,226,227]. The production of the target proteins can be achieved in either prokaryotic or eukaryotic cells, each with their specific advantages and disadvantages [228]. *E. coli* is the most employed microorganism for heterologous protein production due its ease manipulation and cultivation, low costs, and high yields [229]. On the other hand, lack of post-translational modifications, the presence of endotoxin and protein folding distinct from the native protein restrict the application of *E. coli* as antigens factories [230]. Indeed, the maintenance of structural features of flavivirus proteins is a key step in the generation of vaccines able to induce neutralizing antibodies [231,232]. In order to overcome such limitations, technological advances have significantly improved several aspects of heterologous gene expression, including the prediction of solubility by bioinformatics, and they have renewed possibilities for the development of vaccines based on recombinant antigens produced in bacterial cells.

Protein refolding techniques permit recovery of insoluble antigens from inclusion bodies, accumulated in bacterial cells, into soluble proteins. This enables them to generate neutralizing antibodies and protective immunity directed to conformational epitopes exposed on the virus surface both in mice and NHPs (Table 2, Table 3, Table 4 and Table 6). In addition, innovative bioprocessing technologies and the application of scale-up methods for the production of recombinant proteins using *E. coli* have been established [136,229]. There are also novel endotoxin-free strains [233,234], strains that produce chaperones or that promote glycosylation [135]. Another achievement is the development of high-pressure refolding processes that promote antigen folding with characteristics similar to native virus proteins [235,236]. Such advances may reduce both the constraints and the time required for vaccine development and, consequently, accelerate clinical translation.

Subunit vaccines have limited immunogenicity, a phenomenon not restrict to proteins produced in *E. coli* and one that may vary according to the intrinsic characteristics of the antigen [237]. Such vaccines demanded the development of new adjuvants not only to enhance, but also to modulate the antigen-specific immune responses. Other strategy to improve antigen immunogenicity include biotic and abiotic delivery systems, currently a rapidly advancing field of research [237,238,239]. As illustrated, there are several anti-flavivirus vaccine candidates based on proteins produced in *E. coli*, employing different adjuvants or delivery strategies [159,160,161,240,241,242]. These strategies include generation of chimeric proteins able to bind to antigen presenting cells, enhanced induction of neutralizing antibodies, and enhanced T cell responses [159,160,161,240,241,242].

In the case of some anti-flavivirus vaccines, such as DENV vaccines, selection of good antigen/adjuvant combinations is critical, since the ADE phenomenon must be avoided and the induction of balanced cellular and humoral immune responses to the all serotypes is required. This concern is also applied to ZIKV, which has extensive sequence homology with DENV. Several studies have repeatedly demonstrated that proper antigen/adjuvant choices can result in balanced humoral and cellular immune responses and confer protective immunity in mice and NHPs [40,177,178,179,180,181,182,183,219]. Altogether, these findings highlight the potential of these recombinant protein-based flavivirus vaccines toward translation into clinical studies. The development of new flavivirus vaccines is challenging due to the characteristics and incidence of these pathogens and complex economic issues. Clinical trials are very expensive and are usually conducted by pharmaceutical companies with an eye to profit making [243,244]. Tropical diseases such as those caused by flavivirus mainly impact low-income and mid-low-income populations, thus attracting limited interest from private companies. Hence, future investments on the development of alternative and new anti-flavivirus vaccines shall be preferentially tackled by international philanthropic organization and governments of countries more severely affected by these illnesses, such as those in Latin America and Asia. Such decisions are expected to impact the research dealing with production of recombinant proteins at low costs with enhanced safety and efficacy toward generation of protective immunity in humans.

## Figures and Tables

**Figure 1 vaccines-08-00492-f001:**
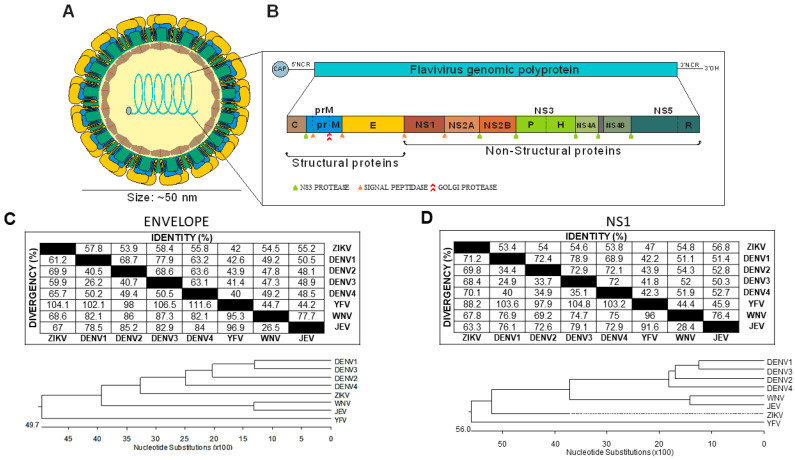
Genetic organization and protein homology among different flaviviruses. (**A**) Schematic representation of the virion structure. (**B**) Genome organization and polyprotein. (**C**,**D**) Homology profiles of Zika virus (ZIKV), Dengue virus (DENV), Yellow Fever virus, (YFV), West Nile virus (WNV), and Japanese encephalitis virus (JEV) were determined using the reference sequences of envelope (E) (**C**) or nonstructural 1 (NS1) (**D**) proteins.

**Table 1 vaccines-08-00492-t001:** Recombinant YFV antigens produced in *Escherichia coli* and other systems.

Antigen	System	Purpose	Soluble	Ref.
EDIII	*E. coli*, pET-15b vector	Structural characterization	No	[34]
EDIII	*E. coli*, pET-20b vector, *pel*B signal sequence	Thermodynamic stability and molecular design	No	[69]
C protein	*E. coli* strain BL21(DE3) RIL, pET-30a vector, lacking 20 C-terminal amino acids	Structural characterization	Yes	[70]
NS2A, NS2B and NS4B	*E. coli* strain C600, *trp*E fusion proteins	Identification of the cleavage sites	No	[11]
NS3	*E. coli* strain W3110, *cro* full-length NS3 fusion and N-terminal truncated NS3	Enzymatic activity characterization	No	[71]
NS2B/NS3	*E. coli* strain M15, pQE30 vector, N-terminal His-tag, hydrophilic core sequence of NS2B linked to NS3 via nonapeptide	Enzymatic activity characterization	Yes	[72]
NS5	HEK293T cells, pIRES/GFP bicistronic mammalian expression vector, His-tagged protein	Phosphorylation characterization	Yes	[73,74]
NS1	*E. coli* strain Lemo21 (DE3), pBT7-N-His vector	Diagnostics	No	[75]
E and NS1 proteins	Vero and *Spodoptera frugiperda* cells	Glycosylation characterization	Yes	[76]
NS1	*E. coli* strain BMH17-18, pUR vector, β-galactosidase fusion	Mice immunization: ↑survival after i.c. challenge	No	[35]
E protein	Transgenic plant (*Nicotiana benthamiana*), N-terminal His-tag	Mice immunization: ↑ neutralizing Ab; ↑ IgG avidity; protection after i.c. challenge Monkey immunization: ↓ Viremia; ↑ neutralizing Ab titers; ↑ IFNy	Yes	[77]

↑ = increased and ↓ = decreased.

**Table 2 vaccines-08-00492-t002:** Strategies for JEV subunit vaccines produced in *E. coli* for preclinical evaluation.

Target	Strategy	Immune Response	Protection	Refolding	Ref.
E protein	Fragments of E protein capable of reacting with neutralizing mAbs	↓Neutralizing Abs	No significant protection	Yes	[99]
E protein	Two fragments of E protein Ea and Eb	Ea: ↓Neutralizing Abs Eb: ↑ Neutralizing Abs	Ea–No significant protection Eb–Partial protection	Ea–Yes Eb–No	[100]
EDIII	Immunization with Freund’s adjuvant or different charged liposomes in mice	Freund’s adjuvant: ↑ Neutralizing Abs Cationic liposomes: ↑ Neutralizing Abs	Freund’s adjuvant: 60% of protection Cationic liposomes: 80% of protection	No	[102]
EDIII	Immunization with Freund’s complete adjuvant Boost with Immunization with Freund’s incomplete adjuvant	↑ Neutralizing Abs	62.5% of protection	No	[36]
NS1	Immunization with Freund’s complete adjuvant Boost with Immunization with Freund’s incomplete	↓Neutralizing Abs	87.5% of protection	No	[36]

↑ = increased and ↓ = decreased.

**Table 3 vaccines-08-00492-t003:** Strategies for WNV subunit vaccines in preclinical assays.

Target	Strategy	Immune Response	Protection	Refolding	Ref.
E protein	Full length E and truncated (80% N-terminal)	Mice: ↑IgG titers. Antibodies recognize WNV-infected cells	Mice: 100% survival. Serum passive transfer 80% survival	No	[123]
	E antigen in a DNA prime and protein boost	Mice: ↑IgG and neutralizing Abs titers. ↑ CD8 + IFNg+ T cells	Mice: 100% survival. ↓ viremia	Yes	[124]
E ectodomain	E ectodomain plus Matrix-M (saponin) adjuvant	Mice: ↑IgG and neutralizing Ab titers	Mice: 100% survival	Yes	[119]
EDIII	EDIII fused with cholera toxin	Mice: ↑IgG, IgM and IgA titers. Complement-mediated killing of serum from immunized mice	ND	No	[125]
EDIII	EDIII genetically fused with flagelin (TLR5 agonist)	Mice: ↑IgG and neutralizing Ab titers	Mice: 100% survival	Yes	[122]
EDIII	EDIII fused with CD40 ligand	Horses: ↑IgG and neutralizing Ab titers	ND	No	[126]
EDIII	EDIII conjugated with VLP of bacteriophage AP205 plus alum	Mice: ↑IgG long-lasting response (>1 year) and neutralizing Ab titers	Mice: 100% survival	Yes	[121]
EDIII	EDIII plus CpG	Mice: ↑IgG titers. Abs recognize WNV-infected cells. ↑T-cell proliferation. ↑cytokines (splenocytes)	Mice: ↑survival of mice receiving WNV premixed with antiserum from immunized mice	No	[127]
EDIII	Comparison EDIII and soluble full-length E	Mice: partially neutralizing Abs. Different immunogens induce different potencies of neutralizing Abs	ND	No	[128]
EDIII	EDIII plus CpG and boosted with oil	Mice: ↑IgG and neutralizing Ab titers	Mice: 80% survival	Yes	[120]
EDIII	EDIII boost after E ectodomain DNA prime	Mice: ↑ IgG and neutralizing Ab titers. Moderate IFN-γ	Mice: 100% survival	Yes	[129]
EDIII	EDIII plus KFE8 peptide hydrogel	Mice: ↑ IgG and neutralizing Ab titers	Mice: 60% survival	Yes	[130]
EDIII	EDIII plus Freund’s adjuvant	Mice: ↑IgG and neutralizing Ab titers	ND	Yes	[131]

ND = not determined; ↑ = increased and ↓ = decreased.

**Table 4 vaccines-08-00492-t004:** Strategies for DENV subunit vaccines produced in *E. coli* for preclinical evaluation.

Target	Strategy	Immune Response	Protection	Refolding	Ref.
E protein	Fusion with P64k protein from *N. meningitidis*	Mice and monkeys: ↑ IgG and neutralizing Ab titers after 4 doses	Mice: 50% survival (DENV2 i.c.) Monkeys: reduction of viremia after challenge (DENV2)	No	[168,169,170]
E protein	Chimeric protein with 11 peptides from DENV1-4 E protein	Mice: ↑ IgG titers; ↑ CD8 and CD4 T cells; splenocytes (↑IFNy, IL-2,IL-4 and IL-17)	ND	Yes	[171]
E protein	DENV2 E protein	Mice: ↑ IgG titers	ND	Yes	[172]
E protein	Fragment of DENV1-4 E proteins fused with maltose binding protein (MBP)	Mice and monkey: ↑ IgG and neutralizing Ab titers	Mice: 80% of survival (i.c. challenge); Monkeys: no protection	No	[173,174,175,176]
EDIII and C proteins	EDIII fused with C proteins from DENV1-4	Monkeys and mice: ↑ IgG and neutralizing Ab titers, ↑ B cells antigen-specific; ↑ IFNy (splenocytes or PBMC); ↑ CD8 e CD8 T cells (IFNy)	Mice: ↑ survival (DENV1-4) Monkeys: ↓ viremia and ↑ survival against DENV1-4	Yes	[177,178,179,180,181,182,183]
EDIII protein	DENV1-4 EDIII protein fused to fliC (*S. typhimurium*)	Mice: ↑ IgG and neutralizing Ab titers (heterologous prime-boost with LATV vaccine); ↓ADE effect	Mice: ↑ survival (DENV1-4)	No	[184]
EDIII protein	DENV1-4 EDIII in tandem (B1234 protein)	Mice: ↑ IgG and neutralizing Ab titers	Mice: ↑ survival (DENV1-4)	Yes	[185]
EDIII protein	DENV2 EDIII fused with pIII coat protein	Mice: ↑ IgG and neutralizing Ab titers	Mice: no protection; ↑ ADE	Yes	[186]
EDIII protein	DENV2 EDIII with different adjuvants (LT1, LTB and Alum)	Mice: ↑ IgG and neutralizing Ab titers	ND	Yes	[187]
EDIII protein	DENV2 EDIII	Mice: ↑ IgG and neutralizing Ab titers	Mice: ↑ survival (DENV2)	Yes	[188]
EDIII protein	DENV1-4 EDIII fused with lipid signal peptide of the lipoprotein Ag473	Mice: ↑ IgG and neutralizing Ab titers; ↑ IgG avidity; ↑ neutralization (PRNT); ↓ ADE in vitro	Mice: ↓ viremia	Yes	[189,190,191,192]
EDIII protein	DENV1 EDIII with different adjuvants (PELC and CpG)	Mice: ↑ IgG and neutralizing Ab titers; ↑ IFNy (ELISPOT)	ND	No	[193]
EDIII protein	DENV1-2 EDIII or DENV3-4 EDIII in tandem	Mice: ↑ IgG and neutralizing Ab titers	Mice: ↑ survival (DENV1-4)	Yes	[194,195]
EDIII protein	DENV2 EDIII with Freund’s adjuvant	Mice: ↑ IgG and neutralizing Ab titers	Mice: ↑ survival (DENV2)	Yes	[196]
EDIII protein	DENV3 EDIII consensus	Mice: ↑ IgG titers; ↑ proliferation of splenocytes; ↑ IFNy and IL-4 (splenocytes)	ND	No	[197]
EDIII protein	DENV1-4 EDIII consensus	Mice: ↑ IgG and neutralizing (DENV1-4) Ab titers	ND	No	[198]
EDIII protein	DENV (1,2) EDIII protein	Mice: ↑ IgG titers	ND	Yes	[199]
EDIII protein	DENV1-4 EDIII	Mice: ↑ IgG and neutralizing Ab titers; ↑ CD4 and CD8 T cells (producing IFNy and IL-2); ↑ IFNy, IL-2, IL-12p40 (splenocytes)	ND	Yes	[200]
EDIII protein	DENV2 EDIII in Chimeric VLP (HBcAg-EDIII-2)	Mice: ↑ IgG and neutralizing Ab titers	ND	Yes	[201]
EDII protein	Fubc protein (peptides from fusion and bc loop regions of EDII fused by link sequence)	Mice: ↑ IgG titers	ND	Yes	[202]
NS1 protein	DENV2 NS1 with LTG33D, Alum or Freund’s adjuvant	Mice: ↑ IgG titers	Mice: ↑ survival (DENV2)	Yes	[37,203]
NS5 protein	DENV2 NS5	Mice: ↑ IgG titers; ↑ IFNy and TNFα (splenocytes)	Mice: ↑ survival (DENV2)	No	[38]
NS3 protein	DENV2 NS3 protein	Mice: ↑ IgG titers; ↑ IFNy (splenocytes)	ND	Yes	[39]
NS3/NS1	DJ NS1 chimera (DENV2 and JEV) and NS3	Mice: ↑ IgG titers;↑ CD4 and CD8 T cells; ↑ CTL responses against NS3	Mice: ↓ viremia; ↓ soluble NS1 levels; ↓ mouse tail bleeding time, and vascular leakage at skin injection sites	Yes	[40]

ND = not determined; ↑ = increased and ↓ = decreased.

**Table 5 vaccines-08-00492-t005:** ZIKV vaccines in clinical trials.

Clinical Trials	Status	Inactivated Virus	Attenuated Virus	DNA-Based	RNA-Based	Viral Vector-Based	Total
Early phase I or phase I	Ongoing	NCT03008122 NCT03343626	-	-	NCT04064905	NCT04033068 NCT04440774 NCT04015648	6
	Completed	NCT02937233 NCT02952833 NCT02963909 NCT03425149	NCT03611946	NCT02840487 NCT02996461	NCT03014089	NCT02996890	9
Phase II	Completed	-	-	NCT03110770	-	-	1

**Table 6 vaccines-08-00492-t006:** Strategies for ZIKV subunit vaccines produced in *E. coli* for preclinical evaluation.

Target	Strategy	Immune Response	Protection	Refolding	Ref.
E protein	90% of the N-terminal whole E protein + alum adjuvant	Mice: ↑IgG and neutralizing Ab titers	Neonatal mice: full protection Pregnant mice: fetus and offspring protection from ZIKV-induced microcephaly	Yes	[216]
E protein	80% of the N-terminal whole E protein + alum adjuvant	Mice: ↑IgG and neutralizing Ab titers; ↑ ZIKV specific T cells; IFNy	Mice: ↑survival	Yes	[218]
E protein	Consensus E protein lacking the whole stem and trans membrane regions + poly (I:C) or CpG ODN adjuvants	Mice: ↑IgG titer; ↑ ZIKV specific T cells; IFNy	ND	Yes	[219]
EDIII protein	Full EDIII sequence + TMG or alum adjuvants	Mice: ↑IgG and neutralizing Ab titers; ↑IFNy, IL-4 and IL-6	ND	Yes	[220]

ND = not determined and ↑ = increased.

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
