# Peer review of "Anti-Flavivirus Vaccines: Review of the Present Situation and Perspectives of Subunit Vaccines Produced in Escherichia coli"

_vaccines, 2020, doi:10.3390/vaccines8030492_

Round 1
Reviewer 1 Report
This manuscript by Araujo et al., systematically reviewed the current situation of anti-Flavivirus vaccine and discussed the perspectives of using recombinant protein-based approaches in future research development and in clinical practicality. The authors summarized and presented the current available anti-Flavivirus vaccine knowledge, which includes aspects regarding each Flavivirus family member: the Yellow Fever virus, the Dengue virus, West Nile virus, Zika virus and the Japanese encephalitis virus. This in-depth summary that puts said viruse vaccine research in one place is needed and helpful for the scientific field. Meanwhile, the authors also presented the challenges and discussed the solutions needed for development of future useful vaccines to each of the above virus. The article is for the most part clearly written and well presented. I have some minor suggestions below:
-P2, line 59, line 62-63, error message meaning?
-Fig 1 fine print unreadable, need higher resolution
-P3, line96 to 100, talks about E protein as a natural candidate for subunit vaccine, which sounds reasonable, before switching to proposals for NS1, NS3 and NS5, the authors should add more discussions and/or current status/progress of using E protein as candidate. If none is done, why not? Then switching to NS1,2,5 would show much smoother transition.
-P3, line 125, should be “……. efficient and QUITE safe”
-P4, line 129, should be “……greater incidence IN elderly”
-P4, line 132, neutralizing antibodies against E protein compared to YF-17D vaccine, although with high titer, but lack of cellular immune response? More clarification is needed here.
-P6, line 217, “……E fragment did not INDUCE neutralizing”
-P6, line 231, line 236 fix error messages
-P16, line 514-517, fix sentence structure:” these vaccines how limitations……”
Author Response
We thank the reviewer for the helpful suggestions. We detail above how we deal with them. We also send the manuscript for a revision by a native English speaker.
-P2, line 59, line 62-63. There was a problem with the reference system and lost some cross-reference citations. We have double-checked in the revised version and eliminated the error messages.
-Fig 1. We did not find how to upload the high-resolution figure. We asked the editors to help us with this issue and sent them the original high-resolution image. We also added a better resolution image saved in another format type (PNG) to the main text.
-P3, line 96 to 100. We changed the paragraph and the text now is the following:
The E protein is a natural candidate for subunit vaccines, since it is on the virus surface and plays a direct role on host cell receptor binding and cell fusion. The ectodomain (soluble N-terminal region) of E monomer has three domains: a beta-barrel domain I (EDI), a finger-like dimerization domain II (EDII) that contains a fusion loop, and an immunoglobulin-like domain III (EDIII), which contains the receptor-binding site and major type-specific neutralization epitopes, consequently, the majority of subunit vaccines candidates uses E protein or EDIII as antigen [33,34]. Nonetheless, cellular immune response can also be protective for flaviviruses and in some cases is required in order to generate robust protection. For these reasons, there are some proposals for subunit vaccines that employ NS1, NS3 and NS5 as vaccine antigens [35–41].
-P3, line 125, should be “……. efficient and QUITE safe”. We have corrected it.
-P4, line 129, should be “……greater incidence IN elderly”. We included the missing preposition.
-P4, line 132. We meant that after YF-17D vaccination most neutralizing antibodies were induced against E protein. We changed the text as following to make it clearer:
"The adaptive immune responses to YFV-17D are fast, robust and durable (usually life-long). Studies show that neutralizing antibodies induced after vaccination with YF-17D target a low number of conserved epitopes in the E protein, and the antibody titers become as high as 30 times the needed amount for protection after vaccination."
-P6, line 217, “……E fragment did not INDUCE neutralizing” We have corrected it.
-P6, line 231, line 236. We have fixed error messages.
-P16, line 514-517. There was an “s” missing, the correct sentence is “these vaccines show limitations...”
Reviewer 2 Report
In this review, authors perform a compendium about recombinant subunit vaccines against the main flavivirus causative of human diseases, as DENV, WNV or ZIKV, among others. They focus specially in subunit vaccines generated in prokaryotic expression systems as E. coli
Overall, this manuscript is well written and organized, going in depth in the field, doing an extensive review and showing the main achievements observed after the immunizations. Minor points should be taken into consideration:
Title: as this work is based on recombinant subunit vaccines in E. coli, the fact of pointing it out in the title is necessary.
Fig 1 is blurred.
16-22 Divide this long sentence in two shorter “Although presently licensed anti-flavivirus vaccines are based on inactivated, attenuated or virus vector vaccines, technological advances on the generation of recombinant antigens with preserved structural and immunological determinants, as well as proposals of multi-epitope vaccines and the discovery of new adjuvants and delivery system that are able to enhance/modulate immune responses, unveil renewed perspectives for the development of recombinant protein-based vaccine formulations to be tested at clinical conditions”.
59,62,63, 231, 236 In these lines “[Error! Bookmark not defined.]” is observed.
94-95 “Structure and function of NS2A, NS4A, and NS4B are still under investigation”. There is information about the role of these proteins:
Dengue NS2A Protein Orchestrates Virus Assembly
Membrane topology and function of dengue virus NS2A protein
NS2A comprises a putative viroporin of Dengue virus 2
Characterization of Dengue Virus NS4A and NS4B Protein Interaction
Determinants of Dengue Virus NS4A Protein Oligomerization
NS4A and NS4B proteins from dengue virus: Membranotropic regions
And so on, and these are only for DENV
Please, complete this information.
187 “Neutralizing antibodies are considered the protection correlate the JEV vaccine”. This sentence sounds rare. “The protection after JEV vaccination correlates with the level of Neutralizing antibodies”
188 that vaccine-induced seroconversion, with neutralization titers of at least 10 in. Please specify units (10)
225-226 “Later, it was reported that the signal peptide of the membrane gene (M) was essential to Ea solubility and induction of protective immunity [74]”. Please, extend this information.
269 “Even after 20 years after the introduction”. Delete the first “after”
277 “Another hurdle for the advancement of efficient WNV vaccines is the epidemiological characteristics of the disease”. Are instead of is
283 “Additionally, the lack of adequate animal models to study WNV infection may difficult
the translation of results to clinical conditions [87]”. 87 reference: Induction of Epitope-Specific Neutralizing Antibodies against West Nile does not show how difficult is the translation of results to clinical conditions. BALB/c mice has been described as a good model for WNV for the study of antiviral, vaccines and pathogenesis
528 Dot lacks between sentences “still needs experimental validation [51,205–207]” and “The production”
Author Response
We thank the reviewer for the kind suggestions and corrections. We describe below how we deal with them. The manuscript was also revised by a native English speaker professional, who has done careful English editing.
We changed the title to point out that the focus are subunit vaccines produced in E. coli. The new title is: Anti-Flavivirus vaccines: review of the present situation and perspectives of subunit vaccines produced in Escherichia coli
Fig 1. As we explained do the reviewer 1, we asked the editors to help us with this issue and sent them the original high-resolution image. We also added a better resolution image saved in another format type (PNG) to the main text.
lines 16-22. We split the text into the following shorter sentences:
"Currently licensed anti-flavivirus vaccines are based on inactivated, attenuated, or virus vector vaccines. However, technological advances in the generation of recombinant antigens with preserved structural and immunological determinants reveal new possibilities for the development of recombinant protein-based vaccine formulations for clinical testing. Furthermore, novel proposals for multi-epitope vaccines and the discovery of new adjuvants and delivery systems that enhance and/or modulate immune responses can pave the way for the development of successful subunit vaccines"
59,62,63, 231, 236 “[Error! Bookmark not defined.]” is observed. As explained to the reviewer 1, there was a problem with the reference system and lost some cross-reference citations. We eliminated the error messages now.
lines 94-95. We completed the information on NS2A, NS4A and NS4B (now lines 88-99). We also included new references and the new text is the following:
“NS2A is small protein reported to be involved in viral RNA replication [11,12], modulation of the host-antiviral interferon response [13–16] and virus particle assembly/secretion [17–19]. NS2B acts as a cofactor to NS3 protease domain, assisting its folding and catalytic activity [8,20]. NS3 has two domains: N-terminal protease and C-terminal helicase. The NS3 protease domain is a chymotrypsin-type serine protease [21]. The NS3 helicase domain presents helicase and nucleoside 5′-triphosphatase activities [22]. NS4A and NS4B have multiple functions involving viral replication and virus-host interactions. The reported functions of NS4A involve endoplasmic reticulum membrane rearrangement [23], participation in virus replication complexes formations [24], autophagy induction to prevent cell death and help viral replication [25] and regulation of NS3 helicase ATPase activity [26]. NS4B is reported to interact with the NS3 helicase domain and dissociate it from single-strand RNA [27]. NS4B can induce the unfolded protein response in the host cells and inhibit interferon (IFN) signaling [16,28,29].”
line 187. We changed the sentence according to the suggestion: "The protection after JEV vaccination correlates with the level of neutralizing antibodies”.
line 188. The number 10 means that the serum was diluted 1:10, and it is a titer value and has no unit. It means the serum dilution, or titer, capable to reduce (neutralize) the number of plaques by 50% compared to the serum free virus. This measurement is denoted as the PRNT50 value and it is the gold standard for detecting and measuring antibodies that can neutralize the viruses. It gives the measure of how effective is a vaccine seroconversion.
We changed the phrase structure to make it clearer: “Several reports demonstrated that a protective status is reached when serum from vaccinated individuals presents neutralization titers in plaque reduction neutralization test (PRNT50) of at least 10”
lines 225-226. We included more details and the text changed to:
“Later, the in vivo expression of Ea in mouse immunized with a DNA vaccine, also found to be non-protective, unless it was preceded by the signal peptide of the gene of M protein, revealing the importance of this peptide to Ea proper folding, immune activity and induction of protection [101]." (now lines 240-243)
line 269. We deleted the repeated word.
line 277. We corrected the verb.
line 283. The discussion of reference [87] brings the following information: "Complicating the issue, many of the initial immunization studies have been performed in mice. Because our studies show that mice generate distinct antibody responses against specific epitopes, some caution may be required in applying mouse vaccination results to humans."
Therefore, we changed the sentence to “the differences between the epitope-specific profile of humans and mice may difficult the translation of mice results to clinical conditions.”
line 528. A dot was included between sentences.
Reviewer 3 Report
The review by Araujo et al. “Anti-Flavivirus vaccines: review of the present situation and perspectives of recombinant protein-based formulations” gives comprehensive analysis of all available and in clinical trials vaccines and debates the vaccine efficacy based on the expression system, demonstrating its potential for the recombinant proteins produced in E. coli.
I do recommend it for publication in Vaccines. However, there are few details that need to be corrected before publication.
The main problem is multiple English errors, which make some sentences difficult to understand. It will be ideal if native English speaking colleague or collaborator can read the manuscript and correct the errors.
Referencing style is inconsistent in the list of references and some errors with referencing are embedded into the manuscript on p2 line59,62; p7 line 231-236
Minor corrections
line 22 of the abstract
“Nonetheless, advances in this field requires high investments” supposed to be “Nonetheless, advances in this field require high investments”
P2 Figure 1 Poor resolution panels, it is impossible to read numbers and labels
Page 3 pine 82-83, 2 parts of this sentence are disconnected
Each flavivirus nonstructural protein has a function in virus replication, and the structure similarity shared by these proteins was recently analyzed.
Page 3 line 101. Reference is missing after “NS1, NS3 and NS5”
Page 4 line 129
These events have greater incidence in elderly.
Page 5 line 173
Japanese encephalitis (JE) is the most important viral encephalitis
Japanese encephalitis (JE) is the most prevalent viral encephalitis
Line 252 reference is missing at the end of the sentence:”
One of the latest outbreaks occurred in 2018 and involved over 2,000 human cases across 15 countries in Europe (ref).
Line 355 And in individuals of 9-45 years old [116,120].
Line 364
364 human challenge trial with an DENV 2 strain [5].
438 the use adjuvants to improve induction
517 as well as to those with more than 60 years of age[34].
as well as to those over 60 years old[34].
523 "subunit vaccines based in"
subunit vaccines based on
524 in different platforms could be economic viable, safe
in different platforms could be economically viable, safe
Author Response
We thank the reviewer for the careful revision of the manuscript. The new version of the manuscript was revised by a native English speaker, who has done careful English editing.
Unfortunately, we remarked the problem with the references only after submission of the first version, but the problem was fixed now.
Abstract, line 22. We corrected the verb.
Page 2, Figure 1. As we explained do reviewers 1 and 2, we asked the editors to help us with this issue and sent them the original high-resolution image and also added a better resolution image saved in another format type (PNG) to the main text.
Page 3 pine 82-83. We split this sentence into two separated ones: “Each nonstructural flavivirus protein has a function in virus replication. The structure similarity shared by these proteins from different flaviviruses was recently analyzed”
Page 3 line 101. The missing references were included
Page 4 line 129. We included the missing preposition.
Page 5 line 173. The word "important" was replaced by "prevalent".
Line 252. The reference was included: Riccardo, F.; Bolici, F.; Fafangel, M.; Jovanovic, V.; Socan, M.; Klepac, P.; Plavsa, D.; Vasic, M.; Bella, A.; Diana, G.; et al. West Nile virus in Europe: after action reviews of preparedness and response to the 2018 transmission season in Italy, Slovenia, Serbia and Greece. Global. Health 2020, 16, 47, doi:10.1186/s12992-020-00568-1.
Line 355. The preposition was changed
Line 364. We changed the sentence to “human challenge model with a DENV 2 strain”
Line 438. We changed the phrase to include the preposition.
Line 517. We changed it accordingly.
Line 523. We corrected the proposition.
Line 524. We corrected the adverb.